# Observation of positronium diffraction

Yugo Nagata [1] ✉, Riki Mikami [1], Nazrene Zafar [2] & Yasuyuki Nagashima [1] ✉

Diffraction of matter waves is a fundamental consequence of quantum mechanics, directly illustrating the core principles of wave-particle duality, quantum superposition, and quantum interference. De Broglie's proposal that particles exhibit wave-like properties has been experimentally confirmed for electrons, neutrons, and composite systems such as helium atoms, molecules and clusters. Here, we report the observation of positronium diffraction using a high-quality, energy-tunable positronium beam transmitted through graphene. Time-of-flight selection and spatially resolved detection reveal a distinct 1st-order diffraction peak at a position consistent with the prediction from matter-wave considerations for positronium. This work provides the direct and definitive evidence of quantum interference in positronium beams and confirms that it behaves as a single quantum entity rather than two independent interfering particles. This groundbreaking experimental milestone marks a major advance in fundamental physics, not only demonstrating positronium's wave nature as a bound lepton-antilepton system but also opening pathways for precision measurements involving positronium.

Positronium (Ps) is the purely leptonic hydrogen-like bound state of an electron and a positron, making it a unique testbed for fundamental physics such as quantum electrodynamics, antimatter interaction and gravity[1]. Ps is the simplest atom composed of equal-mass constituents and, until it self-annihilates (142 ns lifetime for the triplet state), it behaves as a neutral atom in vacuum. Ps can also exist within insulators and may exhibit unique behaviour governed by quantum mechanics. For example, Ps in quartz ($\alpha$-SiO$_2$), as detected using angular correlation of annihilation radiation, can form Bloch states[2,3] at room temperature, while in some alkali halides, this characteristic has been observed at low temperatures[4].

To date, however, there has been no direct observation of Ps wave-like phenomena in free space. While positron matter-wave behaviour has been observed[5,6], its manifestation in exotic composite systems such as Ps has remained largely unexplored. A diffraction or interference experiment with Ps would directly demonstrate its quantum superposition. Such an experiment also addresses whether Ps interferes as a single quantum object or whether its electron and positron constituents act as separate matter waves from observation of the position of the diffraction peak. The distinction is clear from the de Broglie wavelength relation, $\lambda = h/p$, where $h$ is Planck's constant and $p$ is projectile momentum. Because Ps momentum is equal to the sum of the electron and positron momentum, its wavelength is half that of an isolated electron or positron. If the two constituents diffracted separately, the resulting diffraction pattern would exhibit an angular spread approximately twice as large as that expected for a bound quantum system due to the longer de Broglie wavelengths of the individual particles.

Diffraction is a powerful tool in crystal structure analysis. Notably, electrons and, more recently, positrons are used for surface structure determination via reflection high-energy diffraction[7,8]. Since positrons experience a repulsive potential inside a crystal, they undergo total external reflection at grazing incidence, allowing one to probe only the topmost atomic layers. Ps diffraction could similarly be used for non-destructive surface analysis[9]. While previous experimental studies using Ps beams, produced via neutralising slow positron beams in gaseous targets, explored mainly gas phase scattering[10] and specular reflection of Ps from a LiF surface[11], the use of Ps for solid surface diffraction will be an application. Since Ps is electrically neutral, its trajectory is not affected upon incidence at insulator surfaces, which may be charged or generate magnetic fields[12]—a unique advantage over charged particle beams. Though neutron and X-ray beams are similarly insensitive to external fields, Ps offers the added benefit of surface sensitivity, enabling the extraction of information specific to

[1]Department of Physics, Tokyo University of Science, Kagurazaka, Tokyo, Japan. [2]Department of Physics and Astronomy, University College London, London, UK. ✉e-mail: yugo.nagata@rs.tus.ac.jp; ynaga@rs.tus.ac.jp

the outermost atomic layers of solids[9]. Moreover, positrons produced in $\beta$ decay are spin-polarised; Ps beams generated using these positrons should inherit this polarisation[13], opening opportunities for spin-resolved studies of magnetic surfaces. Beyond confirming its quantum properties, the observation of Ps interference opens avenues for fundamental measurements. Owing to its sensitivity, Ps interferometry has been proposed as a method for measuring the gravitational acceleration of antimatter and of purely leptonic systems[14–16]. Though of fundamental interest, no direct gravitational measurement has ever been performed on leptons−even electrons−as gravitational effects are many orders of magnitude weaker than the dominant Coulomb interaction. The matter-wave coherence of dense Ps is also a prerequisite for creating a Bose-Einstein condensate (BEC) of Ps[17,18]. Recent advances in Ps laser cooling[19,20] begin to make it feasible to produce ultracold Ps, an important step towards Ps BEC.

Here we report on the direct observation of Ps diffraction obtained by passing Ps beam at energies of several keV through graphene. This success is due to the fact that the Ps beam used in this study, which is generated by photodetachment of accelerated Ps⁻, possesses sufficient coherence for diffraction observation.

## Results

### Production of a high-quality, energy-tunable Ps beam and preparation of graphene samples

A key requirement for success in Ps diffraction experimentation is the quality of the Ps beam. Here we employed a high-quality, energy-tunable Ps source, developed previously using laser photodetachment of Ps⁻ ions[21]. Compared to Ps beams generated by the neutralisation of slow positrons in gas, this source delivers higher energies (up to 3.3 keV), a narrower energy spread−determined by photoelectron recoil to ~2%−and a reduced angular divergence of less than 0.3 degrees. The fact that Ps beams can be obtained in ultrahigh vacuum is also an important feature to keep the graphene surface clean so that diffraction can be clearly observed. These

characteristics were essential for enabling the observation of Ps diffraction. This beam has already been instrumental in a number of precision measurements, namely: hyperfine structure frequency by motion-induced transition[22]; electron affinity of Ps⁻[23]; anisotropic photodetachment of Ps⁻[24]; and measurements of Ps transmittance through graphene[25].

In this study, positrons emitted from a ²²Na source capsule were moderated by a solid neon moderator and collected in a Surko-type trap[21]. The positrons were then extracted from the trap as a pulsed slow positron beam and incident upon a tungsten film, where a fraction was emitted as Ps⁻ ions from the Na-coated downstream surface. The experimental setup from this point is shown in Fig. 1. The ions were then accelerated to the required energies and photodetached by light pulses from a Nd:YAG laser synchronised with the Ps⁻ pulses. To observe Ps diffraction, the resulting Ps atoms were directed onto a graphene sample. With a lattice constant of 246 pm, graphene is a suitable material with which to perform diffraction experiments in this study, where the Ps de Broglie wavelength is approximately $0.87\,\mathrm{nm}/\sqrt{E/\mathrm{eV}}$. The sample, purchased from EM Japan Co. Ltd., consisted of nominally 2–3 layers of graphene of diameter 2.6 mm, supported by a lacey carbon film on a 300 mesh transmission electron microscopy (TEM) grid to ensure flatness and mechanical rigidity. This was fixed in a stainless steel holder and mounted on a transfer rod. In order to maintain the graphene in optimal condition, it was heated every few hours (see Methods). The component of the Ps beam transmitted through the graphene was detected using a position sensitive detector of active diameter 83 mm consisting of a microchannel plate (MCP) with a delay-line anode (Roentdek, DLD-80). The detector's spatial calibration is described in Methods. The Ps count rates were ~0.02 s⁻¹ at 3.3 keV and ~0.01 s⁻¹ at 2.3 keV. The low rates arise from cumulative losses due to : (i) initial collimation by a 2.0 mm-diameter aperture immediately downstream of the source; (ii) further limitation by the 2.6 mm aperture in the graphene sample holder; (iii) a combined geometric transmittance of ~40% for the lacey carbon

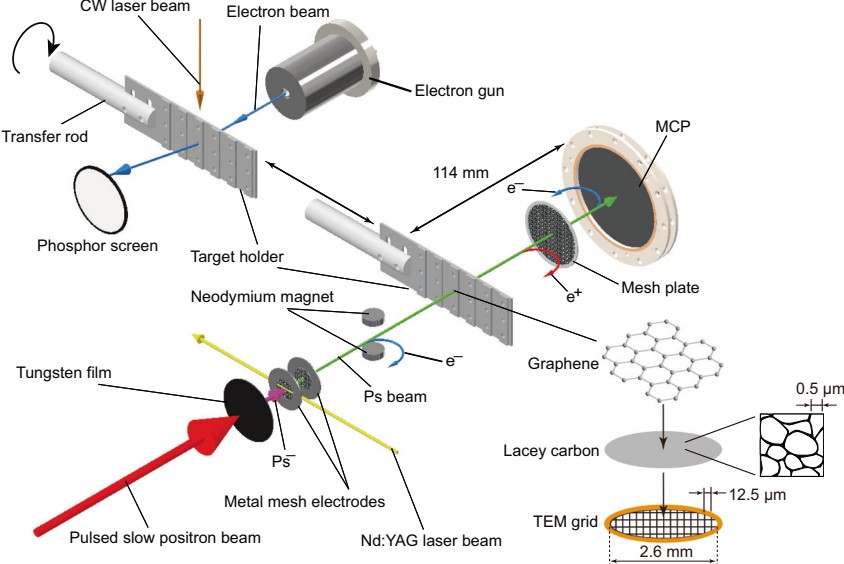

**Fig. 1 | Schematic diagram of the experimental setup (not to scale).** The pulsed slow positron beam from a Surko-type positron trap[31] is injected into a tungsten film at a repetition rate of 50 Hz. The film is Na-coated on the downstream surface to increase the production efficiency of Ps⁻ ions[32–34]. The ions emitted from the surface are accelerated by the voltage difference between the metal mesh electrodes and the film. A Nd:YAG laser beam orthogonal to the beamline and coincident with the positron pulse photodetaches one of the electrons from the ions to produce fast Ps atoms. The Ps beam is collimated by a 2 mm aperture (not shown) and passes through the 2–3 layers of graphene supported by a lacey carbon film

mounted on the target holder to a MCP. Positrons and electrons produced by breakup in the graphene sample (indicated as e⁺ and e⁻ in the diagram, respectively) are repelled by the positively-biased mesh plate in front of the MCP and the negatively-biased MCP front face, respectively. The target holder can be moved inside the vacuum chamber using a transfer rod, and a CW laser beam can be directed onto the graphene for heat treatment. The chamber is also equipped with an electron gun and a phosphor screen to enable checking of the crystal structure of the graphene.

support and 300-mesh Cu grid; and (iv) an additional ~ 50% reduction in flux through the 2–3-layer graphene sample[25].

In the present study, we analysed the data from the MCP detector, which included Ps atoms that had undergone diffraction by the graphene. However, to ensure the integrity of the diffraction signal, contributions from other particles and background noise had to be systematically eliminated. A fraction of the Ps atoms dissociated upon interaction with the graphene target, travelling towards the MCP as electrons and positrons each of ~1.65 keV when the incident Ps energy is 3.3 keV. Electrons thus produced were repelled by the application of a −2 kV electric potential to the front of the MCP, while a mesh plate positioned prior to the MCP, biased at +2 kV, removed the positrons.

Additionally, significant background γ-ray emission arose from multiple sources. Positrons incident on the tungsten film could annihilate without forming Ps, generating prompt γ-rays. Further γ-ray contributions originated from self-annihilation of Ps atoms emitted from the tungsten surface, as well as pickoff annihilation occurring upon collisions with chamber electrodes. Although these high-energy photons interacted inefficiently with the MCP, their substantial numbers posed a risk of spurious signal contamination. Therefore, time-of-flight (TOF) selection, with the timing of the laser irradiation used as the start signal, was employed to discriminate γ-ray events from the Ps

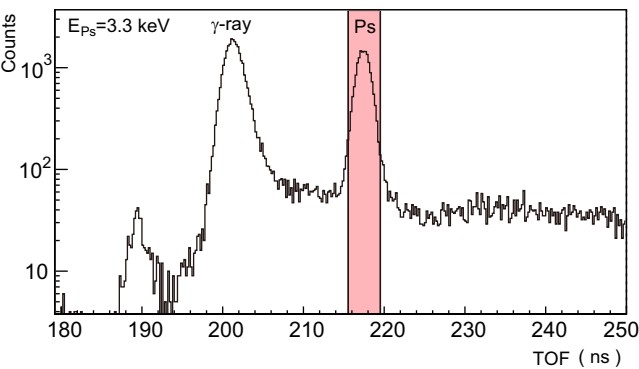

**Fig. 2 | TOF spectrum of the Ps beam at 3.3 keV.** Peaks observed at around 202 ns and 217 ns are due to γ-rays originating upstream and the Ps produced by photodetachment, respectively. The Ps peak (highlighted in red) was observed to be absent when the photodetachment laser was not fired, confirming that this signal is due to Ps atoms[21].

signals. To achieve the statistical precision required for clear identification of diffraction features, data acquisition times were 210 h for the 3.3 keV Ps and 370 h for the subsequent 2.3 keV measurement. Several operational constraints contributed to the overall duration of the experiment: the photodetachment laser system required overnight cooldown; the graphene sample accumulated surface adsorbates, necessitating in-situ cleaning every few hours; and the solid Ne moderator periodically degraded, requiring regeneration every few days.

Figure 2 presents the TOF spectrum recorded for a Ps beam energy of 3.3 keV. Two prominent peaks were observed at ~202 ns and 217 ns. When the Nd:YAG laser pulse was deactivated, the latter peak disappeared[21], confirming that the first and second peaks correspond to γ-ray and Ps events, respectively.

### Experimental results of Ps diffraction

Figure 3a shows the measured event density distribution $\rho(r)$ as a function of radial distance $r$ from the Ps beam centre, for a Ps energy of 3.3 keV, which corresponds to a de Broglie wavelength of 15 pm. The black dots represent events detected by the MCP within the TOF selection window of 215.5–219.5 ns, chosen to isolate Ps events as determined in Fig. 2. A distinct peak is observed near $r$ ~ 8 mm. Arrows at 8.1 mm, 14.0 mm and 16.2 mm indicate the expected positions of the 1st-, 2nd- and 3rd-order Ps diffraction maxima, respectively, calculated using the de Broglie wavelength of Ps and the lattice spacing of graphene. The arrow at 16.2 mm also corresponds to the expected position of the 1st-order diffraction peak that would arise if the electron and positron comprising Ps diffracted independently. Thus, the presence of a prominent peak at 8.1 mm serves as direct evidence for Ps behaving as a single quantum entity, producing interference as a composite particle. In contrast, within the limits of statistical uncertainty in this study, no peaks are seen at either 14.0 mm or 16.2 mm. The absence of a peak at 14.0 mm suggests that 2nd-order Ps diffraction is not observed under the current conditions—unlike the case for electron diffraction, where higher-order peaks typically appear with similar relative intensities (see Methods). Furthermore, the absence of a peak at 16.2 mm confirms that the observed interference pattern is not due to individual diffraction of electrons or positrons, but rather from the coherent diffraction of the bound Ps system.

The Ps diffraction signal is superimposed on a background that includes contributions from γ-ray events originating upstream in the beamline. These γ-ray events were estimated independently using TOF windows of 210–212 ns and 222–224 ns, and are shown as blue triangles

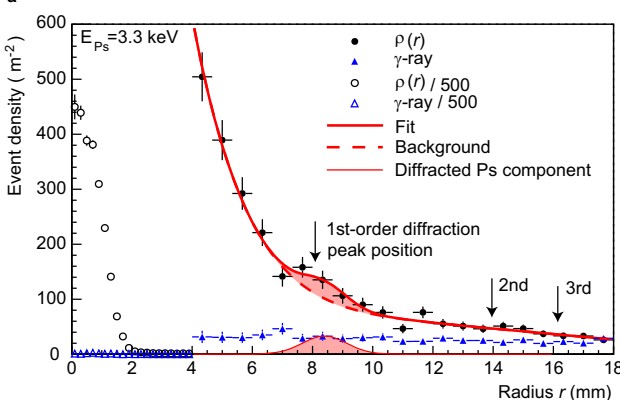

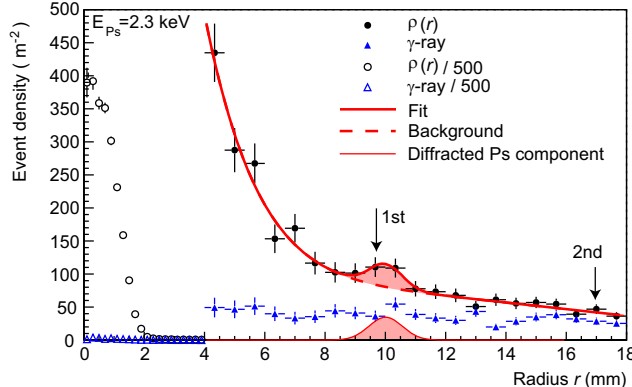

**Fig. 3 | Positronium diffraction results. a** Black dots show the measured event density distribution as a function of radius $r$, $\rho(r)$, for a Ps beam energy of 3.3 keV. Vertical bars represent the statistical uncertainty (standard deviation), and horizontal bars indicate the bin width. The thick red line is a fit to this data using equation (1); the dashed line is a smooth background which is a polynomial fit without the Gaussian; and the thin red line is the Gaussian component obtained by

subtracting the background from $\rho(r)$, which represents the 1st-order diffracted Ps. Blue triangles indicate γ-ray background counts, obtained using TOF selection windows of 210–212 ns and 222–224 ns. For $r < 4$ mm, $\rho(r)$ and γ-ray counts are multiplied by 1/500 and shown as open black circles and open blue triangles, respectively. The shape of $\rho(r)$ in the region $r < 2$ mm indicates transmitted Ps that have not diffracted. **b** Same as in **a** but for a Ps beam energy of 2.3 keV.

in Fig. 3a. For radial distances $r > 4$ mm, the $\gamma$-ray event distribution remains relatively uniform, indicating that it is not significantly dependent on position. Additional background may arise from Ps scattering off C atoms in the lace frame and from reflection and diffraction at the sample holder.

In order to isolate the Ps diffracted component, we fit the following equation, a sum of a Gaussian function representing the 1st-order diffracted Ps and a polynomial representing a smooth background, to the experimental data:

$$\rho(r) \quad = \quad \frac{A}{\sqrt{2\pi\sigma^2}} \exp\left\{ -\frac{(r-\mu)^2}{2\sigma^2} \right\} \\ + c_1|r - r_0|^n + c_2 r + c_3 \tag{1}$$

with parameters of intensity $A$, peak position $\mu$, peak width $\sigma$ and parameters for the background $c_1$, $n$, $c_2$ and $c_3$ applied over the radial range from 4 mm to 18 mm, where $r_0 = 18$ mm. The thick red line in Fig. 3a shows the fitted curve. The reduced chi-squared value of the fit was $\chi^2/\nu = 11.5/14$. The peak position was determined to be $\mu = 8.4 \pm 0.4$ mm, which is consistent with the calculated position of 1st order Ps diffraction $r = 8.1$ mm. The fitted peak width, $\sigma = 0.74 \pm 0.38$ mm, corresponds to a full-width at half-maximum (FWHM) of $1.7 \pm 0.9$ mm, which is consistent with the measured Ps beam diameter of ~2.0 mm (FWHM). As the reduced $\chi^2/\nu$ gave a value slightly less than 1, the possibility of overfitting was considered. To test robustness against the background parameterisation, we repeated the fit with the background parameter $n$ fixed to values about $\pm 25\%$ from that at the best fit of $n = 7.9$, such that $\chi^2/\nu \sim 1$. The inferred peak position changed by $< 4\%$, and the remaining parameters stayed within their uncertainties, indicating that the result is insensitive to the precise background model. The Ps diffraction efficiency at 3.3 keV, $\varepsilon_{Ps}^{3.3keV}$, was determined to be $0.0032 \pm 0.0018$, with the uncertainty calculated from the fitting result. This value is approximately an order of magnitude lower than the electron diffraction efficiency $\varepsilon_e$ of 0.045, measured at 1.6 keV under the equivelocity condition (see Methods).

When the Ps beam energy was decreased to 2.3 keV, the diffraction peak shifted outward, as shown in Fig. 3b. At this energy the de Broglie wavelength is 18 pm. Calculated positions of the 1st and 2nd-order Ps diffraction peaks are indicated by arrows at 9.7 mm and 17.0 mm, respectively. (The 3rd-peak is outside the measurement range in the current experimental setting.) The presence of the peak at 9.7 mm provides direct evidence of Ps diffraction at this Ps energy. No distinct 2nd-order diffraction peak was observed. The experimental data were fitted using the same composite function as described previously, yielding a reduced chi-squared value of $\chi^2/\nu = 9.0/14$, indicating good agreement with the model. The fitted peak position was determined to be $10.0 \pm 0.2$ mm, consistent with the predicted 1st-order Ps diffraction position at 9.7 mm. Again $\chi^2/\nu$ was less than 1, therefore, we repeated the test for robustness by refitting with background parameter $n$ fixed to values $\pm 35\%$ from the best fit of $n = 8.3$, chosen to bring $\chi^2/\nu \sim 1$. The peak position shifted by $< 3\%$, indicating insensitivity to the precise background parameterisation. The Ps diffraction efficiency at 2.3 keV, $\varepsilon_{Ps}^{2.3keV}$, was determined to be $0.0029 \pm 0.0005$, which is of the same order as $\varepsilon_{Ps}^{3.3keV}$. The obtained intensities of Ps diffraction are significantly smaller than that of electrons in the few keV energy region. In single-scattering theories, the diffracted signal scales with the elastic scattering amplitude. For positronium, however, the 1st Born elastic amplitude in a static potential cancels because the electron and positron have equal masses and opposite charges, so the leading non-zero elastic contribution arises only at higher order[26]. By contrast, inelastic channels such as internal excitation, breakup, or pickoff annihilation appear already at first order. These features naturally suppress the coherent elastic signal and provide a consistent explanation for the small Ps diffraction efficiency observed here. While recent studies have revealed

similarities between electron and Ps scattering from molecules at comparable velocities in the range below a few 100 eV[27], these findings have prompted ongoing theoretical investigations[28]. Further theoretical work is needed to interpret the reduced Ps diffraction efficiency observed here and to develop refined models of Ps graphene interactions in the keV energy regime.

The present measurement was conducted using apparatus similar to that used in the work of Mikami et al.[25]. Whereas the previous work[25] required auxiliary measurements to correct for a slight decrease in count rate in order to determine the absolute Ps transmittance, the objective here was a long-duration run that required no such correction. Thus, the data were acquired independently. In this study, we used a graphene thin film consisting of 2–3 layers rather than a monolayer. The diffraction intensity from graphene depends on the number of layers in the sample; consequently, using a monolayer leads to a corresponding reduction in peak intensities. As evident from the present data, halving the diffraction intensity would render the peak comparable to the statistical scatter. To perform the experiment with monolayer graphene, it will therefore be necessary to increase the Ps beam intensity further.

## Discussion

From these observations at Ps beam energies of 3.3 keV and 2.3 keV, the quantum superposition of the Ps matter wave is concluded to have been observed confirming the wave-like behaviour of Ps. This successful observation of Ps diffraction reveals a previously unobserved quantum mechanical property of Ps, that of coherent matter-wave interference. This represents a significant milestone toward future experiments that make use of the wave nature of Ps. To enable applications such as crystal structure analysis, further developments are required, including increasing the Ps beam intensity, reducing its divergence, and minimising its diameter. Additionally, the construction of a Mach-Zehnder interferometer utilising the present technique would be advantageous in realising precision gravity measurements with Ps[16,29].

In conclusion, we report the experimental observation of the 1st-order diffraction peak of Ps through 2–3 layer graphene, providing direct evidence of its wave-like nature. At Ps energies of 3.3 keV and 2.3 keV, the diffraction peaks were measured at positions in agreement with the calculations based on the premise that Ps diffracts as a single quantum entity rather than as two independently interfering electron and positron waves. The diffraction efficiencies were an order of magnitude lower than the electron diffraction efficiency under equivelocity conditions.

This successful observation of Ps diffraction has opened the door to establishing a foundation for Ps-based interferometry with a number of potential applications. Further theoretical and experimental investigations are required to refine our understanding of Ps-graphene interactions and to explore the feasibility of higher-order Ps diffraction under different experimental conditions.

## Method
### Electron diffraction
Prior to conducting the Ps diffraction experiment, electron diffraction patterns were observed in order to evaluate the graphene sample used and to obtain the data for comparison with the Ps diffraction data.

The graphene sample was moved from the Ps beam axis using the transfer rod and repositioned in front of the electron gun, as shown in Fig. 1. The electron diffraction patterns obtained from the sample were observed on a phosphor screen and their profiles were measured using a CCD camera.

Figure 4a shows the CCD profile for an incident electron energy of 6 keV. The characteristic ring structure[30] indicative of 2–3 layer graphene was observed in the obtained pattern. The corresponding radial density distribution is shown in Fig. 4b, where distinct 1st- and 2nd-

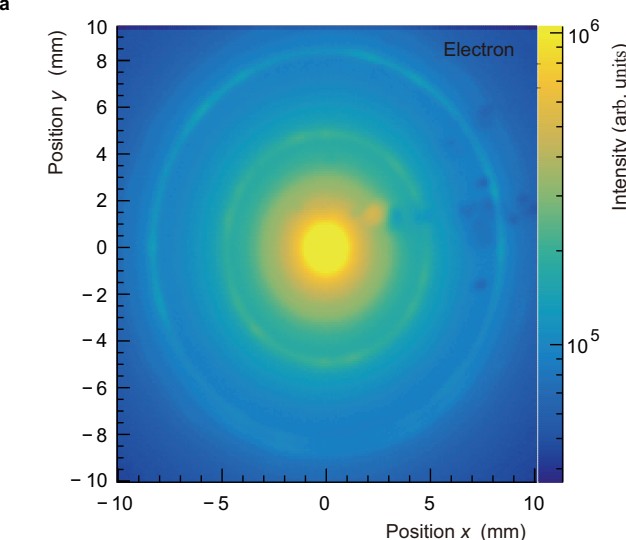

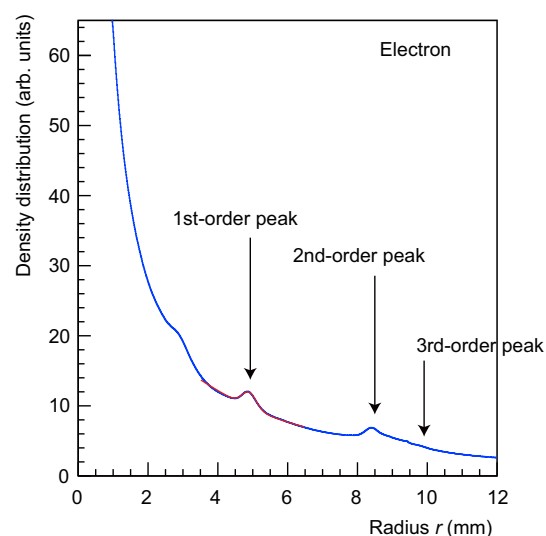

**Fig. 4 | Electron diffraction. a** Electron diffraction pattern for a 6 keV electron beam incident on a 2–3 layer graphene sample obtained using a phosphor screen and a CCD camera. **b** The density distribution for **a** as a function of the radius $r$ from the electron beam centre. Arrows indicate the theoretically calculated positions of 1st-, 2nd- and 3rd-order diffraction peaks. The small shoulder preceding the first diffraction peak is a characteristic of the electron gun, as it appears even in the absence of a sample.

To further optimise the diffraction quality, a heat treatment was applied to the graphene sample using a CW laser with a wavelength of 976 nm and a power density of 70 mW/cm². The laser was activated for 2 min to remove surface adsorbates while the graphene sample was rotated by 90 degrees using the transfer rod. Following this process, $\varepsilon_e$ was noted to increase by a factor of 1.2–1.4 compared to its pre-treatment value. However, it was observed that after ~4 h $\varepsilon_e$ returned to its original level, indicating a gradual reaccumulation of surface contaminants. To maintain optimal diffraction conditions, the heat treatment procedure was repeated every 3–4 hours throughout the Ps diffraction experiment.

Data for comparison with the results of Ps diffraction were measured soon after the heat treatment. For the electrons of energy 1.6 keV, the diffraction efficiency of the 1st-order diffraction peak was 0.045, and the intensity of the 2nd-order peak was of similar magnitude.

### Position calibration of the MCP delay-line detector

The spatial calibration of the MCP delay-line detector was performed using a precision-engineered triangular-lattice array mask mounted directly in front of the MCP. The mask consisted of circular holes with a diameter of 0.25 mm, arranged in a regular triangular grid with a centre-to-centre spacing of 1.09 mm.

For the calibration, positrons from the Surko-type trap were injected directly into the MCP through the mask, producing a well-defined beam profile that replicated the array pattern of the mask. The resulting detected positions were used to determine the MCP detector's spatial response. The $x$- and $y$-coordinate scales of the detector were calibrated using these reference data, with a position resolution of better than 0.2 mm. The mask was removed for the Ps beam experiments.

### Data availability

The source data generated in this study are provided in the Source Data file. Source data are provided with this paper.

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

## Acknowledgements

We would like to thank Paul-Antoine Hervieux, Toma Susi, Philippe Roncin, David Cassidy, Koji Michishio, Luca Chiari, Michael W. J. Bromley and Ayahiko Ichimiya for meaningful discussions on this experiment. This work was supported by JSPS KAKENHI Grants No. JP25H00620, No. JP21H04457, No. JP17H01074.

## Author contributions

Y. Nagat. and R. M. performed the Ps diffraction experiment supervised by Y. Nagas. R. M. and Y. Nagat. prepared graphene samples. Y. Nagat. and R. M. performed the data analysis. The results were discussed by all authors. Y. Nagat. and R. M. wrote the initial manuscript, which was edited together with N. Z. and Y. Nagas., and all authors improved the manuscript.

## Competing interests

The authors declare no competing interests.
