## [Transparent Peer Review file · Nature Communications]

Observation of positronium diffraction

Corresponding Author: Dr Yugo Nagata

Version 0:

Reviewer comments:

Reviewer #1

(Remarks to the Author)

This manuscript presents an experimental demonstration concerning an old and well studied topic, the diffraction of matter waves. The actual specific interest of this current research is that the system under analysis is the positronium (Ps), the atom composed by a positron and an electron. It is very important that the demonstration of a quantum interference is performed on Ps atoms, which are in fact unstable composite particles. On my knowledge, this is the first time that quantum interference experiments concern unstable combinations of leptons and antileptons elementary particles, affected by an annihilation behavior. That this is a groundbreaking experimental advance is correctly claimed in the title and in the abstract of the manuscript.

Coming now to discuss about the core of the manuscript, it is to be noted that the writing is well conducted and extremely clear, starting with a wide discussion on the usefulness of the diffraction and interference phenomena in various research fields, with hint also on recent and future advances, as exposed in literature.

The description of the experimental apparatus is vivid and exhaustive, showing how this difficult experiment is based on a suitable construction of a coherent Ps beam, a technology developed by this research group in recent years.

The analysis of the diffraction signal is performed by separating the background contribution of gamma ray events from the radial density distribution of Ps events, with a suitable fitting model.

Hence the first order diffraction peak results well evident.

Moreover, the radial position of the diffraction peak also demonstrates that the quantum interference involves the whole atom and not the single components, as usefully discussed in the manuscript.

Just a small comment: the diffraction experiment uses graphene samples with 2 or 3 layers as gratings. For the purpose of helping the reader, it can be useful to write explicitly the lattice spacing of the graphene sample, and add some information over the particular samples employed and if it is possible to conduct similar experiments with single layer graphene.

Concluding, in my opinion this manuscript reports about fundamental aspects of quantum wavelike properties on a particular instable system, never analyzed before. Publication is warmly recommended.

Reviewer #2

(Remarks to the Author)

Review Report - Recommendation: Accept after minor corrections.

This manuscript reports the first demonstration of positronium (Ps) beam diffraction on few-layer graphene at kinetic energies of 3.3 keV and 2.3 keV. By employing a positron/electron-repelling MCP combined with a delay-line detector for spatially resolved detection, the authors present unambiguous evidence of Ps wave-like behaviour via de Broglie matter-wave diffraction. The work identifies the first order diffraction peak for Ps, whereas in control measurements with electrons on the same/similar graphene target at scaled energies at least two diffraction rings are observed. The contrast between Ps and electron diffraction efficiency is attributed to a difference in scattering cross-sections, inviting for more detailed studies in the future. This nice experimental demonstration constitutes an important step in establishing Ps as a probe in quantum-optics-like experiments and opens avenues for Ps interferometry. No similar experimental demonstration exists in the published literature to my knowledge, making the work original and impactful.

The experimental execution seems solid. The authors are well known for their excellent Ps- source, the Ps beam preparation through photo-detachment, and the used detection scheme. The identification of the first diffraction maximum is well supported by the quantitative analysis of the data which meets the expected standards in the field. The absence of higher-order peaks in Ps compared with electron data is surprising, but it is not explicitly attributed to the low Ps scattering cross-section they found – see my detailed comments. The electron data shows an unexplained first peak that is not labeled “first peak” – again see my detailed comments.

The manuscript is clearly written, logically structured, and quite accessible to readers in the field, sometimes lacking a few details in my opinion. Prior work on Ps beams and on electron diffraction is appropriately cited, and the novelty of the present contribution is evident. The references appear generally appropriate.

Detailed comments:

p1, left, l8 Maybe add for clarity that Ps also forms at the surfaces of solids.

p1, right, l14 Since your matter wavelength depends on the velocity as well, I would appreciate to see a value for your Ps wavelength just for information.

p2, left, l43 Since you list the Ps beam properties, why not also state your current number of Ps atoms in the beam – it will also fit later with your presented measurement time.

p2, right, l2 Can you give an example for why it is an important feature? Or do you mean it is important for your experiment?

p2, right, l10 Could you highlight the experimental differences to the measurement you did in [36] Ps transmittance through graphene? I presume the choice of 2-3 layers of graphene in this work is based on the results of that publication?

p3, left, l29 I would remove that last sentence: “As a result [...] it took two years”; 600h measurement time and maintenance is a tough experimental challenge, in particular to keep systematic drifts under control. It takes the time it takes.

p3, right, l2 Please mention in values the lattice spacings you used for the calculation, and if not before, at latest here report the used matter wavelengths. My over the thumb calculation for Ps at 3.3keV finds the first maximum at 8.1 mm, the second at 14.1mm the third at 16.3mm.

p4, left, l4 and l31 The reduced chi-squared is consistently <1 , I assume your model is overfitting a little. Is it possible to fix n in the fit?

p4, left, l20 Please state the new matter wavelength as well.

p4, right, l21 Can you use your found efficiencies as an explanation why you cannot see the second order diffraction signal? If this is not the issue, what is your explanation that you do not see it?

p9, caption, g4 What is the smoothed background? Is this the outcome of the polynomial fit without the Gaussian? Please clarify.

General: Can you specify the type of your delay-line detector or give a reference in the methods section?

p10, left, l17 At what distance from the graphene, and at what energy (later you write that for Fig. 4b it was 1.6keGV not 6keV). Please clarify.

p10, left, l29 Again the smooth background is a polynomial fit?

p10, left, l34 Typo: taking _into_ account

Fig. 4b There is an unexplained peak at around 2.5mm, whereas at 3rd order nothing is to be seen. I would like to know why there is this first peak in your electron data and where it comes from. it seems significant enough to not be able to leave it without explanation.

Reviewer #3

(Remarks to the Author)

Dear Editor,

I have enjoyed reading the article “First observation of positronium diffraction” by Nagata et al. This is a very hard experiment and will be of interest due to the subject matter being anti-matter. The carrying out of the experiment and the methods used are incontrovertible. However, in the wide field of anti-matter diffraction there is little that can be called new, except that it has now been done with Positronium. This paper is probably out of date since the publication of “Diffraction of atomic matter waves through a 2D crystal” by Kanitz et al., but it should definitely be published in a widely read journal like Physical Review.

The manuscript presents an interesting topic, namely wave particle duality, which is one of the key cruxes of quantum mechanics. The wave nature of electrons was empirically verified by the Davisson-Germer experiment in 1927. Wide interest in the topic has led to similar confirmations from matter constituents (neutrons, helium, larger molecules) scattering off of structures, potentials, crystals, etc. as noted in the manuscript. It has even been shown that anti-matter (positron) has similar particle-wave nature in 2019 (doi/10.1126/sciadv.aav7610).

The production of Ps from photo-ionizing the Positronium-minus ion is well understood, as are the detection methods. A great deal of time is needed in collecting statistically significant data, and as such, the results are not ambiguous and the conclusion incontrovertible. Details or a reference for the delay line detector should be given. I enjoyed reading this manuscript, but I am left with the following thought: is there a newfound technique or analysis developed through the experimental methods, or new knowledge resulting from this measurement, namely that -much like the hydrogen atom, a bare electron or a positron- Ps also has a wavelike nature as a bound state.

Detailed comments

- Line 5 through 8 on page one (left), the authors describe Ps as an atom and go on to say that Ps behaves as a neutral atom; however, Ps acts like a neutral atom, however when Ps atoms move through a gas they scatter like a lone electron (Brawley et al. 2010 doi/10.1126/science.1192322]
- Throughout the paper, the authors claim that the observation of Ps interference opens new avenues for fundamental measurements (see page 2 left line 25 or page 4 (right) line12). The authors should continue by highlighting some potential measurements. The authors could consider spin sensitive measurements with polarized positron production. Even so, how exactly does this Ps diffraction measurement open a new avenue, and what new avenue is there that the reader would have not known before?
- On page 4, line 45-46 (left) the authors note that their "findings have prompted ongoing theoretical investigations" but I wonder if there is any disagreement. It seems that the authors did not measure anything surprising.
- In figure 3, is the increased signal between the 2nd and 3rd peaks explained by the detection of gamma rays?
- In figure 4, there is a prominent peak before the first electron diffraction peak. This should be addressed.
- Lastly, I am curious as to why the diffraction efficiency was orders of magnitude lower than the electron diffraction (see page 4 line 29-31 right). Maybe detailing the variance of graphene samples (e.g. lattice spacing) might be helpful.

I thoroughly enjoyed reading this manuscript and I am happy that the authors were able to complete such a difficult measurement. Even though the results are neither ground breaking nor surprising, I believe that following slight revisions, in particular highlighting potential new experiments, this manuscript should be published in a widely read journal such as Nature.

Reviewer #4

(Remarks to the Author)

Version 1:

Reviewer comments:

Reviewer #1

(Remarks to the Author)

Second report

This manuscript was beautifully revised by the Authors following the concerns of mine and of other referees, solving any remaining questions and adding useful details on the experiment and its motivation.

I am absolutely convinced that this is a fundamental experiment demonstrating quantum interference on Ps atoms, which is a unique lepton-antilepton system in which the components have the same role with equal or opposite physical properties.

The experiment also demonstrates that the quantum interference involves the whole atom and not the single components, an issue of great importance for any theoretical analysis and studies on antimatter.

Hence, I strongly support publication.

I only noticed one small issue: the attribute "First" in the title ("First observation of Ps diffraction" in the previous version of the title) and in other sentences within the manuscript was removed, surely (I believe) by the Authors. In my opinion this research is really deserving this attribute, which illuminates and attract interest on this fundamental topic, but of course I accept the

will of the authors.

Reviewer #2

(Remarks to the Author)

I thank the authors for thoroughly addressing all my points in a satisfying manner and adding the requested details to the manuscript. I think it is very clear now. I have no more questions and fully recommend publication.

Reviewer #3

(Remarks to the Author)

Reviewer #4

(Remarks to the Author)

Reply to referee's comments

We are pleased to note the favourable comments provided by the four referees. We thank them for their careful reading of our manuscript and for their valuable constructive feedback. We have addressed all points in a detailed, point-by-point response and revised the manuscript accordingly.

Sincerely,

Yugo Nagata, Riki Mikami, Nazrene Zafar, Yasuyuki Nagashima

Reviewer #1

This manuscript presents an experimental demonstration concerning an old and well studied topic, the diffraction of matter waves. The actual specific interest of this current research is that the system under analysis is the positronium (Ps), the atom composed by a positron and an electron. It is very important that the demonstration of a quantum interference is performed on Ps atoms, which are in fact unstable composite particles. On my knowledge, this is the first time that quantum interference experiments concern unstable combinations of leptons and antileptons elementary particles, affected by an annihilation behavior. That this is a groundbreaking experimental advance is correctly claimed in the title and in the abstract of the manuscript.

Coming now to discuss about the core of the manuscript, it is to be noted that the writing is well conducted and extremely clear, starting with a wide discussion on the usefulness of the diffraction and interference phenomena in various research fields, with hint also on recent and future advances, as exposed in literature. The description of the experimental apparatus is vivid and exhaustive, showing how this difficult experiment is based on a suitable construction of a coherent Ps beam, a technology developed by this research group in recent years.

The analysis of the diffraction signal is performed by separating the background contribution of gamma ray events from the radial density distribution of Ps events, with a suitable fitting model. Hence the first order diffraction peak results well evident. Moreover, the radial position of the diffraction peak also demonstrates that the quantum interference involves the whole atom and not the single components, as usefully discussed in the manuscript. Just a small comment: the diffraction experiment uses graphene samples with 2 or 3 layers as gratings. For the purpose of helping the reader, it can be useful to write explicitly the lattice spacing of the graphene sample, and add some information over the particular samples employed and if it is possible to conduct

similar experiments with single layer graphene.

Concluding, in my opinion this manuscript reports about fundamental aspects of quantum wavelike properties on a particular instable system, never analyzed before. Publication is warmly recommended.

Reply: We thank the referee for the favourable assessment, and for recognising the significance of this work for fundamental physics.

Comment 1) *For the purpose of helping the reader, it can be useful to write explicitly the lattice spacing of the graphene sample, and add some information over the particular samples employed and*

Reply: Lattice constant of graphene is 246 pm.

In page 2, right, line 37, we inserted the lattice spacing of 246 pm.

Comment 2) *if it is possible to conduct similar experiments with single layer graphene.*

Reply: The diffraction intensity of graphene depends on the number of layers of graphene in the sample used. Consequently, employing monolayer graphene results in a corresponding decrease in the intensities of the diffraction peaks. As evident from the present data, if the diffraction intensity halves, the peak would become obscured by statistical scatter. Indeed, measurements using monolayer graphene yielded data where the diffraction peak was not clearly discernible.

We added the following sentence in page 5, left, line 16 as,

“In this study, we used a graphene thin film consisting of 2–3 layers rather than a monolayer. The diffraction intensity from graphene depends on the number of layers in the sample; consequently, using a monolayer leads to a corresponding reduction in peak intensities. As evident from the present data, halving the diffraction intensity would render the peak comparable to the statistical scatter. To perform the experiment with monolayer graphene, it will therefore be necessary to increase the Ps beam intensity further.”

Reviewer #2

This manuscript reports the first demonstration of positronium (Ps) beam diffraction on few-layer graphene at kinetic energies of 3.3 keV and 2.3 keV. By employing a positron/electron-repelling MCP combined with a delay-line detector for spatially resolved detection, the authors present unambiguous evidence of Ps wave-like behaviour via de Broglie matter-wave diffraction. The work identifies the first order diffraction peak for Ps, whereas in control measurements with electrons on the same/similar graphene target at scaled energies at least two diffraction rings are observed. The contrast between Ps and electron diffraction efficiency is attributed to a difference in scattering cross-sections, inviting for more detailed studies in the future. This nice experimental demonstration constitutes an important step in establishing Ps as a probe in quantum-optics-like experiments and opens avenues for Ps interferometry. No similar experimental demonstration exists in the published literature to my knowledge, making the work original and impactful.

The experimental execution seems solid. The authors are well known for their excellent Ps- source, the Ps beam preparation through photo-detachment, and the used detection scheme. The identification of the first diffraction maximum is well supported by the quantitative analysis of the data which meets the expected standards in the field. The absence of higher-order peaks in Ps compared with electron data is surprising, but it is not explicitly attributed to the low Ps scattering cross-section they found –see my detailed comments. The electron data shows an unexplained first peak that is not labeled “first peak” – again see my detailed comments. The manuscript is clearly written, logically structured, and quite accessible to readers in the field, sometimes lacking a few details in my opinion. Prior work on Ps beams and on electron diffraction is appropriately cited, and the novelty of the present contribution is evident. The references appear generally appropriate.

Reply: We thank the referee for the favourable assessment, and for recognising the significance of this work for fundamental physics.

Comment 1) *p1, left, 18 Maybe add for clarity that Ps also forms at the surfaces of solids.*

Reply: Thank you for the helpful suggestion. Our intent here was to give a flavour of the states in which Ps exists, rather than the details of its production.

In page 1, left, line 7, we have rewritten this sentence as “it behaves as a neutral atom in vacuum. Ps can also exist within insulators and may exhibit unique behaviour governed by quantum mechanics.”

Comment 2) *p1, right, 114 Since your matter wavelength depends on the velocity as well, I would appreciate to see a value for your Ps wavelength just for information.*

Reply: In page 2, right, line 36, we added “With a lattice constant of 246 pm, graphene is a suitable material with which to perform diffraction experiments in this study, where the Ps de Broglie wavelength is approximately $0.87 \text{ nm}/\sqrt{E/\text{eV}}$. “

Comment 3) *p2, left, 143 Since you list the Ps beam properties, why not also state your current number of Ps atoms in the beam – it will also fit later with your presented measurement time.*

Reply: The Ps beam count rates were 0.02 cps at 3.3 keV and 0.01 cps at 2.3 keV. These low values are due to: (i) the Ps beam being collimated through a 2 mm diameter aperture immediately after the generation, (ii) being further collimated by the holder of the graphene sample having a diameter of 2.6 mm, (iii) the combined geometrical transmittance of the lacey carbon and Cu grid being approximately 40% and (iv) the number of Ps being reduced by about half in 2–3-layer graphene (Mikami et al., Eur. Phys. J. D 77, 205 (2024)). Please note that as the MCP detection efficiency has not been measured, we confine the metric to count rate rather than the Ps beam intensity.

In page 3, left, line 9, we added “The Ps count rates were $\sim 0.02 \text{ s}^{-1}$ at 3.3 keV and $\sim 0.01 \text{ s}^{-1}$ at 2.3 keV. The low rates arise from cumulative losses due to : (i) initial collimation by a 2.0 mm-diameter aperture immediately downstream of the source; (ii) further limitation by the 2.6 mm aperture in the graphene sample holder; (iii) a combined geometric transmittance of $\sim 40\%$ for the lacey carbon support and 300-mesh Cu grid; and (iv) an additional $\sim 50\%$ reduction in flux through the 2–3-layer graphene sample [30].”

Comment 4) *p2, right, 12 Can you give an example for why it is an important feature? Or do you mean it is important for your experiment?*

Reply:

The vacuum is important for this experiment and, more generally, for Ps diffraction from 2D crystals. Ultra-high vacuum is required to suppress the adsorption of contaminants on the graphene surface; otherwise adsorbates attenuate or blur the diffraction peaks and alter the surface potential, compromising the measurement.”

In page 2, right, line 15, we added as “to keep the graphene surface clean so that diffraction can be clearly observed.”

Comment 5) *p2, right, 110 Could you highlight the experimental differences to the measurement you did in [36] Ps transmittance through graphene? I presume the choice of 2-3 layers of graphene in this work is based on the results of that publication?*

Reply: The Ps beam intensity gradually decreases over time during the day. For this reason, in the transmittance experiment [36], it is necessary to measure data under various conditions of 1-layer, 2-layer, 3-5 layer, 6-8 layer graphene and no sample alternately in a short period of time in order to normalise by no sample data correctly. On the other hand, we measured only under the one condition of the 2-3 layer graphene in the diffraction experiment as it did not require us to normalize the data in the same way. We chose 2–3-layer graphene for the diffraction experiment informed by [36]: it provides significantly higher diffracted intensity than a monolayer (while keeping multiple scattering and attenuation modest), and it is mechanically more robust as a freestanding membrane. Using a monolayer would reduce the peak heights sufficiently that, at the current beam intensity, the first-order ring would be comparable to statistical noise.”

In page 5, left, line 10, we inserted the following sentence as, “The present measurement was conducted using apparatus similar to [30]. Whereas [30] required auxiliary measurements to correct for a slight decrease in count rate in order to determine the absolute Ps transmittance, the objective here was a long-duration run that required no such correction. Thus, the data were acquired independently. In this study, we used a graphene thin film consisting of 2–3 layers rather than a monolayer. The diffraction intensity from graphene depends on the number of layers in the sample; consequently, using a monolayer leads to a corresponding reduction in peak intensities. As evident from the present data, halving the diffraction intensity would render the peak comparable to the statistical scatter. To perform the experiment with monolayer graphene, it will therefore be necessary to increase the Ps beam intensity further.”

Comment 6) *p3, left, 129 I would remove that last sentence: “As a result [...] it took two years”; 600h measurement time and maintenance is a tough experimental challenge, in particular to keep systematic drifts under control. It takes the time it takes.*

Reply: We removed “As a result [...] it took two years”.

Comment 7) *p3, right, l2 Please mention in values the lattice spacings you used for the calculation, and if not before, at latest here report the used matter wavelengths. My over the thumb calculation for Ps at 3.3keV finds the first maximum at 8.1 mm, the second at 14.1mm the third at 16.3mm.*

Reply: In page 2, right, line 36, we added “With a lattice constant of 246 pm, graphene is a suitable material with which to perform diffraction experiments in this study, where the Ps de Broglie wavelength is approximately $0.87 \text{ nm} \sqrt{E/\text{eV}}$.”

We also added “which corresponds to a de Broglie wavelength of 15pm.” in page 3, right, line 20.

Comment 8) *p4, left, l4 and l31 The reduced chi-squared is consistently <1, I assume your model is overfitting a little. Is it possible to fix n in the fit?*

Reply:

We thank the referee for raising this point. To test whether the reduced chi-squared values (χ^2/ν) below unity indicate over-fitting, we repeated the fits while fixing the background-shape parameter n to a series of values around the best-fit, and examined both χ^2/ν and the fitted first-order peak position.

Figures (a) and (c) show results for the 3.3 keV data. In (a), the circles give χ^2/ν (with $\nu = 15$) as a function of fixed n ; the square marks the value obtained when n is allowed to float (best-fit $n=7.9$; in the manuscript this appears as $\chi^2/\nu = 11.5/14$ —see correction below). As n is moved away from 7.9, χ^2/ν increases and exceeds 1 at both ends of the scan, as expected if the floating- n solution is preferred. In (c), the corresponding first-order peak position shows only a weak dependence on n , varying by $< 4\%$ across the scanned range.

Figures (b) and (d) present the same test for 2.3 keV. The behaviour is similar: χ^2/ν rises toward and above unity as n is displaced from the best-fit value (8.3), while the peak position changes by $< 3\%$. These checks indicate that, although χ^2/ν is somewhat below 1 when n floats, the physical observables are insensitive to the precise background parameterisation, and there is no evidence that over-fitting biases the peak location.

We also correct a typographical error in the degrees of freedom: $\nu = 14$ (not 13).

Accordingly, the values quoted in the manuscript should read $\chi^2/\nu = 11.5/14$ (p. 4, left, l. 17) and $\chi^2/\nu = 9.0/14$ (p. 4, right, l. 13).

In page 4, left, line 33, we have added the following clarifying sentences: “As the reduced χ^2/ν gave a value slightly less than 1, the possibility of overfitting was considered. To test robustness against the background parameterisation, we repeated the fit with the background parameter n fixed to values about $\pm 25\%$ from that at the best-fit of $n=7.9$ such that $\chi^2/\nu \sim 1$. The inferred peak position changed by $<4\%$, and the remaining parameters stayed within their uncertainties, indicating that the result is insensitive to the precise background model.”

In page 4, right, line 25, we added the following sentence as,

“Again χ^2/ν was less than 1, therefore we repeated the test for robustness by refitting with background parameter n fixed to values $\pm 35\%$ from the best-fit of $n=8.3$, chosen to bring $\chi^2/\nu \approx 1$. The peak position shifted by $<3\%$, indicating insensitivity to the precise background parameterisation.”

Comment 9) *p4, left, l20 Please state the new matter wavelength as well.*

Reply:

In page 4, right, line 11, we added “At this energy, the de Broglie wavelength was 18 pm.”.

Comment 10) *p4, right, l21 Can you use your found efficiencies as an explanation why you cannot see the second order diffraction signal? If this is not the issue, what is your explanation that you do not see it?*

Reply:

Thank you for your insightful comment. On its own, the measured first-order Ps diffraction efficiency does not determine the second-order efficiency, because the relative order intensities depend on the (unknown) Ps elastic amplitudes. However, if we assume—as in electron diffraction—that the second-to-first order intensity ratio is ~ 0.7 (see Shevitski et al., Phys. Rev. B 87, 045417 (2013)), then the event density at the ring position scales as the diffraction efficiency divided by the area of $2\pi r dr$. Therefore, the event density for the 2nd order diffraction decreases by $0.7 * 8.1/14.1=0.4$, so the expected second-order peak height would be only $\sim 40\%$ of the first-order peak in the $\rho(r)$ plot. Because the γ -ray background is approximately uniform for $r > 4$ mm, this reduced signal is comparable to the background fluctuations at $r \approx 14$ mm, making the second-order peak statistically difficult to resolve with the present data set. Any additional suppression of Ps elastic scattering relative to electrons would further decrease the visibility of the second order.

Comment 11) *p9, caption, g4 What is the smoothed background? Is this the outcome of the polynomial fit without the Gaussian? Please clarify.*

Reply: Yes, the smoothed background is a polynomial fit without the Gaussian.

In page 11, caption, g4, we added as "which is a polynomial fit without the Gaussian".

Comment 12) *General: Can you specify the type of your delay-line detector or give a reference in the methods section?*

Reply:

We used a DLD80 (Roentdek). In page 3, left, line 7, we added "(Roentdek, DLD-80)".

Comment 13) *p10, left, l17 At what distance from the graphene, and at what energy (later you write that for Fig. 4b it was 1.6keV not 6keV). Please clarify.*

Reply: Thank you for pointing this out. Fig. 4a and 4b both correspond to the 6 keV electron data taken with the same sample and geometry. We chose to present the figures because the electron-gun brightness at lower energies was insufficient to produce as clean a pattern and peak integration.

The 1.6 keV value mentioned in the text refers to a separate measurement used solely

for the equal-velocity comparison with 3.3 keV Ps; it is not the dataset shown in Fig. 4.

Comment 14) *p10, left, 129 Again the smooth background is a polynomial fit?*

Reply: Yes, we used a quadratic polynomial for the smoothed background.

In page 6, left, 5, we added as “using a Gaussian function with a smooth background of a quadratic polynomial.”

Comment 15) *p10, left, 134 Typo: taking_into_account*

Reply: Thank you very much. We fixed it.

Comment 16) *Fig. 4b There is an unexplained peak at around 2.5mm, whereas at 3rd order nothing is to be seen. I would like to know why there is this first peak in your electron data and where it comes from. it seems significant enough to not be able to leave it without explanation.*

Reply: Thank you for drawing attention to these features.

A third-order ring is expected but its intensity is weak for our thickness and electron energy, and its radius lies close to that of the second order. In the present geometry these two orders partially overlap, so the third order is not resolved as a separate maximum.

The small shoulder at ~2.5 mm is not a lattice reflection. It is an instrumental artefact of the electron gun, as the same feature appears in calibration runs without any sample inserted.

We added the following in the figure 4 caption: “The small shoulder preceding the first diffraction peak is a characteristic of the electron gun, as it appears even in the absence of a sample.”

Reviewer #3 (Remarks to the Author):

Dear Editor,

I have enjoyed reading the article “First observation of positronium diffraction” by Nagata et al. This is a very hard experiment and will be of interest due to the subject matter being anti-matter. The carrying out of the experiment and the methods used are incontrovertible. However, in the wide field of anti-matter diffraction there is little that can be called new, except that it has now been done with Positronium. This paper is probably out of date since the publication of “Diffraction of atomic matter waves through a 2D crystal” by Kanitz et al., but it should definitely be published in a widely read journal like Physical Review.

Reply:

Thank you for your encouraging comments. We appreciate your recognition of the experimental difficulty and your positive assessment of our methodology.

Regarding novelty, our central contribution is not merely that “it has now been done with positronium,” but that we provide the first direct demonstration of quantum interference for a lepton–antilepton bound atom. Positronium is qualitatively different from ordinary atoms used in prior diffraction studies (including Kanitz et al.): it is neutral antimatter, unstable (142 ns triplet lifetime), and its elastic amplitude in first-Born approximation cancels due to the equal masses and opposite charges of its constituents. This leads to a strongly suppressed coherent signal and makes diffraction of Ps uniquely challenging. Observing a clear first-order peak therefore establishes that Ps interferes as a single quantum entity, rather than as independently diffracting electron and positron—an issue with no analogue in standard atomic beams. It also provides the first quantitative constraint on Ps–graphene elastic scattering in the keV regime, relevant for Ps interferometry and gravity tests.

Concerning Kanitz et al., we agree it is an important advance in **ordinary-matter atom** diffraction through 2D crystals. The subject of **our** paper is the interference of matter waves in a bound system of **particles and antiparticles**, rather than of ordinary matter atoms. This has truly fundamental importance and requires experimental verification

Our experimental concept is easily understood and the relationship between particles and antiparticles is the subject that may interest many readers. We are confident that this paper is suitable to Nature Communications.

Reply: We thank the referee for their favourable assessment, and for recognising the significance of this work for fundamental physics.

Comment 1) *Line 5 through 8 on page one (left), the authors describe Ps as an atom and go on to say that Ps behaves as a neutral atom; however, Ps acts like a neutral atom, however when Ps atoms move through a gas they scatter like a lone electron (Brawley et al. 2010 doi/10.1126/science.1192322)*

Reply: Thank you for this helpful clarification. We agree that, at low collision energies, Ps–molecule scattering cross sections closely follow those for electron impact at equal projectile velocity (Brawley et al., Science 330, 789–792, 2010). This important observation does not imply that Ps ceases to be an atom; it reflects similarities in the interaction at low energies and does not contradict the fact that Ps is a neutral bound e^+e^- atom whose centre-of-mass wave can diffract as a single quantum entity. The sentence on p.1 was intended to emphasise Ps’s neutrality and field insensitivity in vacuum, and not to make a statement about gas-phase scattering. To avoid ambiguity we have revised it to: “it behaves as a neutral atom in vacuum.” in page 1, left, line 7.

Comment 2) *Throughout the paper, the authors claim that the observation of Ps interference opens new avenues for fundamental measurements (see page 2 left line 25 or page 4 (right) line12). The authors should continue by highlighting some potential measurements. The authors could consider spin sensitive measurements with polarized positron production.*

Reply: Thank you—we agree and have expanded revised the text to note the feasibility of spin-resolved surface studies using polarised Ps derived from beta decay positrons. In page 2, left, line 24, we added as “Moreover, positrons produced in β decay are spin-polarised; Ps beams generated using these positrons should inherit this polarisation [13], opening opportunities for spin-resolved studies of magnetic surfaces.”

Comment 3) *Even so, how exactly does this Ps diffraction measurement open a new avenue, and what new avenue is there that the reader would have not known before?*

Reply: Thank you for your comment. First, the feasibility of coherent Ps interference is an experimental question—non-trivial for a short-lived, annihilating, lepton–antilepton atom. Our measurement demonstrates that Ps does diffract as a single quantum entity, thereby enabling interferometric schemes that were previously only proposed.

Second, the data show that the elastic (coherent) signal is strongly suppressed relative to electrons at equal velocity, implying different scattering amplitudes for Ps–crystal interactions. This quantitative insight is actionable: it sets requirements for beam intensity, sample thickness, and background control in planned Ps interferometers. With these two points, the work opens concrete paths that a reader would not have been able to assume a priori: (i) spin-resolved Ps interferometry (using polarised β -decay positrons) for magnetic-surface studies; (ii) Mach–Zehnder Ps interferometers for precision tests of gravity and short-range forces; and (iii) surface metrology with neutral antimatter, exploiting Ps’s intrinsic surface sensitivity in UHV. Future designs must account for the observed Ps specific scattering properties revealed by the present diffraction measurement.

Comment 4) *On page 4, line 45-46 (left) the authors note that their "findings have prompted ongoing theoretical investigations" but I wonder if there is any disagreement. It seems that the authors did not measure anything surprising.*

Reply: Thank you—we did not intend to imply any disagreement with established theory. Rather, the relevant theory is incomplete: most Ps–scattering work treats molecules below a few hundred eV, whereas keV-energy Ps interacting with crystalline solids (and Ps diffraction from 2D lattices) lacks a predictive framework. In this regime the first-Born elastic amplitude cancels, so the diffracted intensity depends on higher-order terms and on the competition with inelastic channels—issues that have not yet been quantified.

Comment 5) *In figure 3, is the increased signal between the 2nd and 3rd peaks explained by the detection of gamma rays?*

Reply: Thank you for the question. In Fig. 3, the apparent excess between the nominal second- and third-order radii lies within the 1–2 σ scatter of neighbouring bins. Our TOF sidebands indicate a nearly flat γ -ray background for $r > 4$ mm; together with a small contribution from the undiffracted/forward-scattered Ps tail, this accounts for the observed level. An unresolved second-order component cannot be excluded, but with the present statistics it cannot be distinguished from the background.

Comment 6) In figure 4, there is a prominent peak before the first electron diffraction peak. This should be addressed.

Reply: Thank you for pointing this out. This feature is not a lattice reflection: it appears with the sample removed, so it originates from the incident electron beam (gun/optics) rather than graphene. We have noted this explicitly in the text and caption (Fig. 4).

We added the following in the figure 4 caption: “The small shoulder preceding the first diffraction peak is a characteristic of the electron gun, as it appears even in the absence of a sample.”

Comment 7) *Lastly, I am curious as to why the diffraction efficiency was orders of magnitude lower than the electron diffraction (see page 4 line 29-31 right). Maybe detailing*

the variance of graphene samples (e.g. lattice spacing) might be helpful.

Reply: The key difference from electrons is the elastic scattering amplitude. In the keV regime one normally applies first-Born (kinematic) diffraction theory. For positronium in a static potential, however, the first-Born elastic amplitude cancels because the electron and positron have equal masses and opposite charges; the leading non-zero elastic term therefore arises only at second order (or higher) [H. Ray, Phys. Lett. A 252, 316 (1999)]. At the same time, inelastic channels (break-up, internal excitation, pickoff annihilation) enter already at first order, depleting the coherent flux. These two effects naturally suppress the diffracted intensity for Ps by orders of magnitude relative to electrons at equal velocity.

Regarding sample variance: the graphene lattice constant is essentially fixed (246 pm) and does not vary appreciably between our membranes; it sets the ring positions but does not explain the efficiency difference. What can affect the intensity are thickness (number of layers), cleanliness, and support-grid open fraction. We therefore used 2–3-layer graphene (as informed by Ref. [36]) to improve the signal while keeping multiple scattering modest.

In page 4, right, line 36, we added the following sentences as,

“In single-scattering theories, the diffracted signal scales with the elastic scattering amplitude. For positronium, however, the 1st Born elastic amplitude in a static potential cancels because the electron and positron have equal masses and opposite charges, so the leading non-zero elastic contribution arises only at higher order [31]. By contrast, inelastic channels such as internal

excitation, breakup, or pickoff annihilation appear already at first order. These features naturally suppress the coherent elastic signal and provide a consistent explanation for the small Ps diffraction efficiency observed here.”

In page 2, right, line 37, we inserted the lattice spacing of 246 pm.